# Evaluation of Ethanol Extract of *Moringa oleifera* Lam. as Acaricide against *Oligonychus punicae* Hirst (Trombidiformes: Tetranychidae)

**DOI:** 10.3390/insects12050476

**Published:** 2021-05-20

**Authors:** Rapucel Tonantzin Quetzalli Heinz-Castro, Roberto Arredondo-Valdés, Salvador Ordaz-Silva, Heriberto Méndez-Cortés, Agustín Hernández-Juárez, Julio César Chacón-Hernández

**Affiliations:** 1Faculty of Agronomy and Veterinary, Universidad Autónoma de San Luis Potosí, Soledad de Graciano Sánchez 78321, San Luis Potosí, Mexico; rapucel.heinz@uaslp.mx (R.T.Q.H.-C.); heriberto.mendez@uaslp.mx (H.M.-C.); 2Faculty of Chemical Science, Universidad Autónoma de Coahuila, Saltillo 25280, Coahuila, Mexico; r-arredondo@uadec.edu.mx; 3Faculty of Business and Engineering San Quintín, Universidad Autónoma de Baja California, San Quitín 22930, Baja California, Mexico; salvador.ordaz.silva@uabc.edu.mx; 4Parasitology Department, Universidad Autónoma Agraria Antonio Narro, Saltillo 25315, Coahuila, Mexico; chinoahj14@hotmail.com; 5Institute of Applied Ecology, Universidad Autónoma de Tamaulipas, Victoria City 87019, Tamaulipas, Mexico

**Keywords:** avocado bronze mite, oviposition, hatched eggs, damage, feed intake, mortality, residuality

## Abstract

**Simple Summary:**

Avocado bronze mite (ABM), *Oligonychus punicae* Hirst (Acari: Tetranychidae) is one of the most economically important pests in avocado cultivars. Its feeding causes major damage, defoliation and fruit abortion. Control measures of ABM are performed mainly through the use of synthetic acaricides. Alternative control strategies for ABM with a low environmental impact are necessary. The aim of this research was to assess the effect of different concentrations (0.1, 0.5, 1, 5, 10, 15 and 20% (*v*/*v*)) of ethanolic extract from *Moringa oleifera* leaves against adult ABM females. Mites treated with 0.1 and 20% (*v*/*v*) of the extract showed mortality of 0.00% and 46.67%, 6.67% and 86.67%, 13.70% and 96.67%, at 24, 48 and 72 h, respectively, compared to the control group. The oviposition and eggs hatch, as well as ABM feeding rates, depended on the extract concentration, which led to a reduction in the growth rate. The *M. oleifera* leaves ethanolic extract has potential to control *O. punicae*.

**Abstract:**

Tetranychidae family is a major group of mites causing serious damage in agricultural, vegetable and ornamental crops. Avocado bronze mite (ABM), *Oligonychus punicae* Hirst (Acari: Tetranychidae) causes major crop damage, defoliation and fruit abortion. At present, the control of this mite depends mainly on agrochemicals. Therefore it is necessary to find alternatives to synthetic pesticides that can help minimize environmental impact and health risks for the consumers. The aim of this research was to assess the effect of different concentrations (0.1, 0.5, 1, 5, 10, 15 and 20% (*v*/*v*)) of ethanolic extract of *Moringa oleifera* leaves against adult ABM females. Mites treated with 0.1 and 20% (*v*/*v*) of the extract showed mortality of 0.00% and 46.67%, 6.67% and 86.67%, 13.70% and 96.67%, at 24, 48 and 72 h, as compared to the control treatment, respectively. The number of eggs laid and hatch, as well as ABM feeding rates, depended on the extract concentration, which led to a reduction in the growth rate. *M. oleifera* leaf ethanolic extract has potential to control *O. punicae*.

## 1. Introduction

Tetranychidae include around 1326 identified species. They are a major group of mites, which feed from agricultural, vegetable and ornamental crops. More than one hundred species are considered pests and around 10 of them are important [1]. Most of these pest mites belong to two genera, *Tetranychus* Dufour and *Oligonychus* Berlese [2]. Two hundred and thirteen (213) *Oligonychus* species have been found feeding on perennial grasses, ornamental bushes and trees [3]. Avocado bronze mite (ABM), *Oligonychus punicae* Hirst (Acari: Tetranychidae), is native to Mexico and feeds on 34 plant species. It is distributed in Neotropical countries such as Brazil, Colombia, Costa Rica, Cuba, El Salvador, Honduras, Mexico, Nicaragua, Panama and Venezuela [1,3,4]. In Mexico, ABM is a major pest affecting several avocado cultivars (*Persea americana* L.). High populations of these mites cause defoliation, leading to sunscalds in fruits, causing fruit drops, due to fruit-setting abortion, providing lower yields [4,5]. Chemical control is the main ABM management method. However, ABM’s short life cycle and high fertility cause them to develop resistance to acaricides very quickly [4,5]. Furthermore, synthetic pesticide residues can cause harmful effects on human health and have a high environmental impact. For these reasons, there is an urgent need to develop safe, ecological techniques [6].

Moringa tree, *Moringa oleifera* Lam. (Moringaceae), grows in tropical and subtropical countries and all the tree parts are used for different purposes, including human and animal health, as well as traditional medicine. Different parts of *M. oleifera* have been studied, in particular the leaves, because of its wide range of applications and properties. Literature contains reports stating that Moringa leaves have several bioactive compounds such as vitamins, carotenoids, polyphenols, phenolic acids, flavonoids, alkaloids, glucosinolates, isothiocyanates, tannins, saponins, oxalates and phytates [7]. Several researchers have reported that *M. oleifera* has pest control properties. The leaf powder from *M. oleifera* has anti-egg-laying activity over *Callosobruchus maculatus* F. (Coleoptera: Chrysomelidae) and *Tribolium castaneum* Herbst (Coleoptera: Tenebrionidae) [8]. Likewise, *M. oleifera* oil has anti-feeding properties over *Spodoptera littoralis* Boisd larvae (Lepidoptera: Noctuidae) [9] and *S. frugiperda* Walker (Lepidoptera: Noctuidae) [10]. Furthermore, the lectins in *M. oleifera* seeds have larvicidal effect over the developing instars of *Aedes aegypti* mosquito L. (Diptera: Culicidae) [11] and over the Mediterranean flour moth *Ephestia kuehniella* Zeller (Lepidoptera: Pyralidae) [12]. The aim of this research was to assess the effects of ethanolic extract of *M. oleifera* leaves on mortality, food intake, oviposition, hatching and growth rate of *O. punicae* at different concentrations, since such effects may serve as useful basis for a compound in *O. punicae*’s integrated management program

## 2. Materials and Methods

### 2.1. Colony

The ABM females used in this research work came from a colony of plants infested by *Pithecellobium dulce* (Roxb.) Benth. (Fabaceae) in Ciudad Victoria, Tamaulipas (23°44′38.4″ N, 99°9′57.599″ O, 329 msnm) [13]. In order to increase ABM population, adult females and males were placed on bean plants (*Phaseolus vulgaris* L. (Fabaceae)), under greenhouse conditions 29 ± 4 °C and 60 ± 15% relative humidity (RH).

### 2.2. Preparation of the Plant Material and the Extract

We collected visibly clean leaves of *M. oleifera* in the Applied Ecology Institute at Universidad Autonoma de Tamaulipas. The leaves were dried in a conventional oven (Quincy lab, Chicago, IL, USA model 20GCE-LT) at 60 °C for three days, until obtaining a consistent weight. The sample was grounded (Miller CUISINART, Stamford, CT, USA, model DBM-8) until forming particles of 1 mm [14]. The powder was stored in dark bottles at ambient temperature in preparation for the extraction.

Fourteen grams of the dry homogenized *M. oleifera* leaf powder was blended with 200 mL of absolute ethanol at ambient temperature. This was done during three days with the aid of a stirrer in total darkness. The blend was filtered with Whatman No. 1. Filter paper. The extract was placed in a rotary evaporator (IKA-RV 10 digital V, Staufen Baden-Wurtemberg, Germany) to evaporate the solvent, at temperatures lower than 40 °C and under reduced pressure. Finally, the remaining ethanol was evaporated placing the flask on the kiln drier until obtaining a consistent weight (three days) [15]. The extract was scraped off and stored into Eppendorf tubes and kept in a freezer at −10 °C, before conducting the bioassays.

### 2.3. Phytochemical Extract Analysis

We performed the ethanolic extract analysis of *M. oleifera* in order to do the qualitative detection tests of phytochemicals. The test included alkaloids (Dragendorff & Sonheschain reagent); phenols (Iron Chloride test FeCl3); carbohydrates (Molisch reagent); carotenoids (H_2_SO_4_ and FeCl_3_ reagents); coumarins (Erlich reagent); flavonoids (Shinoda reagent and NaOH at 1%); free reducing sugar (Fehling and Benedict reagent); cyanogenic glycosides (Grignard reagent); purines (HCl test); quinones (NH_4_OH and H_2_SO_4_ reagents by anthraquinone, and Börntraguer test by benzoquinone); saponins (foam test, Bouchard reagent for steroid saponins and Rosenthaler reagent); terpenoids (Ac_2_O reagent); soluble starch (KOH and H_2_SO_4_ test) and tannins (FeCl_3_ reagent and ferrocyanide) [16,17,18].

### 2.4. Experimental Design

We used the sand technique described by Ahmadi [19], with a slightly modification. Bean discs of 2.5 cm in diameter were cut and placed on water-soaked cotton, with the underside facing up. A disc was placed inside a 5 cm wide Petri dish, each disc had 10 adult ABM females with one day age. The experiment was done under laboratory conditions at 27 ± 1 °C, 70–80% relative humidity (RH) and a photoperiod of 12:12 (light:darkness). The bean leaf discs were split at random in eight groups: a control group and seven treatment groups, one per each extract concentration. A bean leaf disc was the replicate. We had three replicates per group. Ten mite females were placed on each bean leaf disc and they were sprayed two times (0.7 ± 0.1 mL per spray) with each concentration. A manual sprayer (Truper^®^ Model 14687 Ciudad de Victoria, Mexico) was used to apply *M. oleifera* extract at different concentrations [0.1, 0.5, 1, 5, 10, 15, 20% *v*/*v* (ethanol/water)]. The control treatment was sprayed with distilled water only. We recorded the number of dead mites at 24, 48 and 72 h. Mites with ataxia (active, apparently messy movement) were considered dead, as well as mites lying on their backs, legs up, or without moving. We also counted the number of eggs laid and we observed the damage caused by ABM. ABM eggs’ hatching determined the residual effect, based on the number of eggs laid at 24 h. In average, ABM eggs hatch between 4.4 and 4.7 days 27 ± 2 °C with a photoperiod of 12:12 h light:dark and 80 ± 10% RH, on different host plants [20]. Therefore, five days after applying the concentrations, we recorded the number of eggs hatched. The eggs that did not hatch within that time were considered non-viable. The counts and feeding observations were conducted with a dissection microscope (UNICO Stereo and Zoom Microscopes ZM180, Princeton, NJ, USA).

### 2.5. Oligonychus punicae Mortality Essay

We corrected the mortality data using Abbott’s formula on the control group [21].
MC = [(%M_T_ − %M_C_)/(100 − %M_C_)] × 100 (1)
where %M_T_ is the mortality percentage in the treated group and %M_C_ is the mortality percentage in the control group.

### 2.6. Oligonychus punicae Oviposition and Hatched Eggs Essay

We used Kramer and Mulla [22] formula to determine the oviposition activity percentage (OAP) in each concentration.
OAP = [(N_T_ − N_C_)/(N_T_ + N_C_ )] × 100(2)
where N_C_ is the number of eggs in the control group and N_T_ is the number of eggs in the treated group with the extract. The percentage values ranged between +100 and −100. The positive values indicate that we observed more eggs laid in the treatment than in the control group (showing that the extract stimulated egg-laying activity). In contrast, more eggs laid in the control group than in the treatment resulted in a negative OAP, indicating that the extract inhibited the egg-laying activity.

In order to measure the residual effect of the extract on ABM’s egg hatching, we used the criterion of Kramer and Mulla [22] formula. The negative values indicate that there were a larger number of eggs hatched in the control group than in the group of treated mites. This shows that the residual effect of the extract inhibited egg-hatching, or that the eggs take longer to develop. The positive values indicate that there is a larger number of eggs hatched in the group of treated mites than in the control group, showing that the extract stimulated hatching in ABM eggs, or that the development time is less than 4.4 or 4.7 days [20].

### 2.7. Oligonychus punicae Anti-Feeding

We assessed the anti-feeding effect through the inhibition percentage of feed intake by ABM in the bean leaf discs, as compared to the control treatment. At 72 h, we classified the symptoms observed in the bean leaf discs according to an ordinal scale developed by Hussey and Parr [23] and Nachman and Zemek [24]. We converted them into percentages: 0 = 0% damage (with no feeding damage), 1 = 1–20%, 2 = 21–40%, 3 = 41–60%, 4 = 61–80% and 5 = 81–100% of feeding damage (dense marks, or wilting, after eating all the bean disc). We used the criterion of Kramer and Mulla [22] formula to measure the females’ food intake. The positive values show that there was more damage in the treatment group than in the control group, indicating that the treatment promoted feeding. The negative values represent more severe damage to the control group than to the treatment group, indicating that feeding was inhibited in the treatment group [25].

### 2.8. Oligonychus punicae Growth Population

We used the growth rate (r_i_) as a parameter to determine the effect of the extract on ABM population.
r_i_ = ((N_f_/N_0_))/∆t (3)
where N_0_ is the initial number of individuals (10 adult females by replicate); N_f_ is the final number of individuals (adult surviving females plus the eggs laid at the end of the bioassay) and Δt, which is the number of days elapsed from the beginning until the end of the bioassay (equal to 3 days). Positive values of r_i_ indicate a growing population. The negative values indicate a declining population, and r_i_ = 0 indicates a stable population [26,27].

### 2.9. Statistical Analysis

Mortality percentage, the number of laid eggs, eggs hatched, the percentage of feeding damage and the growth rate were statistically analyzed using the variance analysis (ANOVA) and the means were separated by post-hoc Tukey’s multiple range comparison test (*p* ≤ 0.05). Probit analysis was used to estimate the lethal concentration (LC_50(90)_), causing 50(90)% mortality in *O. punicae* population, with confidence intervals of 95% (CI95) [28]. SAS/STAT program was used in every analysis [29].

## 3. Results

### 3.1. Phytochemical

The ethanolic extract of *M. oleifera* leaves has several groups of secondary metabolites, such as phenols, alkaloids, flavonoids, tannins, saponins, carbohydrates and quinones (Table 1).

### 3.2. Acaricidal Effect of M. oleifera Extract on Oligonychus punicae

#### 3.2.1. Mortality

We evaluated the effect of seven concentrations of ethanolic extract of *M. oleifera* leaves against adult females of ABM (Table 2). At 24, 48 and 72 h after extract application there was a significant effect on the number of ABM female (F = 64.81; df = 6, 14; *p* < 0.0001; F = 119.56; df = 6, 14; *p* < 0.0001; F = 163.82; df = 6, 14; *p* < 0.0001), respectively. At 24 h, mortality ranged between 0.00% (0.1% (*v*/*v*)) and 70.00% (20.0 (*v*/*v*)), by 48 h, between 3.33 (0.1% (*v*/*v*)) and 83.33% (20.0 (*v*/*v*)), and by 72 h between 10.37 (0.1% (*v*/*v*)) and 96.67% (20.0 (*v*/*v*)), compared to the control treatment (Table 2). Lethal concentration LC_50(90)_ was 7.99(15.68)% (*v*/*v*); meaning, that only 7.99(15.68)% (*v*/*v*) are required to kill 50(90)% of *O. punicae*’s population (Table 2). The results showed that the percentage of *O. punicae* mortality increased as the extract concentration increased.

#### 3.2.2. Oviposition

The number of *O. punicae* eggs laid, varied significantly at 24, 48 and 72 h (F = 940.69; df = 7, 16; *p* < 0.0001; F = 1360.56; df = 7, 16; *p* < 0.0001; F = 2668.89; df = 7, 16; *p* < 0.0001), respectively. Mites treated with 0.1% and 20% (*v*/*v*) inhibited egg-laying by 2.54% and 94.69%, 2.92% and 94.45%, 3.63% and 95.26% at 24, 48 and 72 h, respectively, as compared to the control treatment. The growth rate of the ABM was also statistically significant among the treatments, versus the control group (F = 1311.75; gl = 7, 16; *p* < 0.0001) (Table 3) and it decreased according to the concentration.

#### 3.2.3. Eggs Viability

The percentage of eggs hatched at the fifth day was statistically significant (F = 881.36; df = 7, 16; *p* < 0.0001). The extract concentrations of 0.1% and 20.0% caused a residual effect, leading to a decrease in the number of eggs hatched, ranging between 14.90% and 100.00%, as compared to the control treatment (Table 4).

#### 3.2.4. Food Intake

The damage percentage was significantly different among the treatments, at 24, 48 and 72 h (F = 70.66; df = 7, 16; *p* < 0.0001; F = 121.95; df = 7, 16; *p* < 0.0001; F = 220.57; df = 7, 16; *p* < 0.0001), respectively. Feeding activity of ABM females treated with 0.1% to 20% (*v*/*v*) of *M. oleifera* extract was inhibited by 4.37% to 80.16%, 5.12% to 78.95% and 4.71% to 83.57% at 24, 48 and 72 h, respectively (Table 5).

## 4. Discussion

Secondary metabolites such as steroids, alkaloids, terpenoids, phenolic compounds and essential oils present in more than 2000 plant species have insecticide properties. In developed countries, a few by-products of these plants are used as botanical insecticides [30,31]. Botanical insecticides represent 1% of the global insecticide market [30,31]. In this research work, the phytochemical analysis of ethanolic extract of *M. oleifera* leaves showed the presence of several secondary metabolites, such as flavonoids, alkaloids and cyanogenic glycosides. These bioactive compounds can be useful to pest management strategies [32]. Secondary metabolites (flavonoids, alkaloids and cyanogenic glycosides) have insecticidal effects; inhibiting development as well as egg-laying and feeding rates of mites. They can also be repellent, toxic and can have anti-feeding, attractant and killing effects on several species of insects and herbivores [32].

There is no reference in literature regarding the use of *M. oleifera* leaves ethanolic extract for the control of *O. punicae* or any other mite species. However, the effects of other plant extracts on *Oligonychus* spp. have been studied and those results are similar to the results of this work. This research shows that *O. punicae* mortality rate increases in relation higher extract concentrations. Similarly, egg-hatching, egg-laying, growth rate and feeding rate decrease as concentration increases. Roy et al. [33], reported the mortality of *Oligonychus coffeae* Nietner (Tetranychidae: Acarina) at concentrations of 2%, 4%, 6%, 8% and 10% (*v*/*v*) of *Polygonum hydropiper* L. (Polygonaceae) aqueous extract. The mortality percentage increased at 24 h (26.58% to 80.07%), 48 h (26.58% to 86.41%) and 72 h (26.58% to 89.92%), as compared to the control (0.00% at 24, 48 and 72 h). Furthermore, Sarmah et al. [30] documented that aqueous extracts of *Acorus calamus* L. (Araceae), *Clerodendrun infortunatum* L. (Verbenaceae), *Xanthium strumarium* L. (Compositae) and *Polygonum hydropiper* L. (Polygonaceae) increased the mortality of *O. coffeae* at 24 h (6.4% to 87.7%; 23.0% to 100.0%; 15.6% to 87.2%; 12.8% to 77.7%), 48 h (6.4% to 87.7%; 23.0% to 100.0%; 15.6 to 87.2; 12.8% to 77.7%) and 72 h (6.4% to 88.7%; 23.0% to 100.0%; 15.6% to 91.8%; 12.8% to 84.2%) at concentrations ranging between 2.5% to 10% (*w*/*v*), respectively. In addition, Fetoh and Al-Shammery [34] found that the concentrations (10, 100, 1000, 10,000 and 100,000 ppm) of the ethanolic extract from Demsisa (*Ambrosia maritimal* L. (Compositae)), Duranta, (*Duranta plumeria* L. (Verbenaceae)) and Cumin (*Cuminum cyminum* L. (Labiaceae)) caused mortality in *O. afrasiaticus* McGregor (39.00% to 93.00%; 33.33% to 69.00%; 12.00% to 64.67%) at 24 h. Mamun et al. [35] reported that extracts from *P. hydropiper*, *X. strumarium*, *Datura metel* L. (Solanaceae), *Lantana camara* L. (Verbenaceae), *Swietenia mahagoni* (L.) Jacq. (Meliaceae) and *Azadirachta indica* A. Juss. (Meliaceae) cause mortality in *O. coffeae*. Roobakkumar et al. [36] found that the aqueous extracts of Pongam and Garlic seeds, as well as Neem seeds caused mortality of 100% and 90% of *O. coffeae* mites, respectively. Roy and Mukhopadhyay [37] documented a mortality rate ranging from 0.00% to 90.60% of *O. coffeae* adults treated with 1 to 10% (*w*/*v*) of seed aqueous extract of *Melia azedarach* L. (Meliaceae) at 72 h.

In this research, the number of eggs laid by *O. punicae* females lowered as *M. oleifera* extract concentration increased. Roy et al. [31] documented that *P. hydropiper* aqueous extract reduces the egg-laying activity (1.8 to 1.4 eggs/female) of *O. coffeae* in all the concentrations (2 to 10% (*v*/*v*)) at 24 h, as compared to the control (3.4 eggs/female). This effect became stronger with time, reducing oviposition from 1.4 eggs/female to zero eggs/female, at concentrations of 8 and 10% in 24 to 96 h. In addition, Roobakkumar et al. [36] reported that at 5% concentration (*w*/*v*), the seed aqueous extracts of Neem, Pongam and Garlic, inhibited by 96, 80 and 88% the egg-laying capacity of *O. coffeae*, as compared to the control at 96 h. While Fetoh and Al-Shammery [34] found that the ethanolic extract of Demsisa, Duranta and Cumin decrease egg-laying activity in *O. afrasiaticus* at 24 h (1.3, 2.20, 3.30), 48 h (0.50, 1.65, 3.70) and 72 h (0.30, 6.95, 7.75 eggs/female), in contrast with the control (6.10, 6.35 and 11.66 eggs/female). Likewise, Roy and Mukhopadhyay [37] reported that the concentrations (1–10%) of *M. azedarachun* seed aqueous extract reduced by 40% to 54% the number of eggs laid by every female of *O. coffeae* per day, compared to the control.

In this research, the percentage of *O. punicae* hatched eggs decreased according to the concentration of *M. oleifera* extract, indicating that all concentrations have a residual effect, since they lowered the viability of eggs laid ranged between 14.90 (0.1% (*v*/*v*)) to 100.00% (20.0% (*v*/*v*)), as compared to the control group. In this regard, Dimetry et al. [38] and Dimetry et al. [39] mentioned that secondary metabolites can directly influence over the feminine ovaries, or it can be due to the contact of the mite’s cuticle with a substance present in the botanical insecticide that can alter the production of pheromones, what causes the eggs not to hatch. Fetoh and Al-Shammery [34] mentioned that chemicals present in plants could block the micropyle region of the egg, impairing gas exchange and leading to the embryo’s death inside the egg. Hosny et al. [40] mentioned that adult mite females exposed to discs treated with acaricides, could be subject to partial or temporary sterilization, resulting in a lower number of eggs laid per female, per day; as well as a lower number of viable eggs, in comparison to the control. Therefore, we may conclude that *M. oleifera* causes these two effects and can be characterized by the sterilizing effect.

The ethanolic extract of *M. oleifera* inhibited ABM food intake since the first 24 h, indicating an inverse relation between the extract concentration and the anti-feeding effect (damage). Fetoh and Al-Shammery [34] documented that concentrations at 47.6, 1102 and 8433.2 ppm of the ethanolic extracts from Demsisa, Duranta and Cumin deter impaired by 95.33% and 97.80%; 66.67% and 97.80%; 55.33% and 93.33% *O. afrasiaticus* feeding rate on bean plants (*P. vulgaris*) at 24, 48 and 72 h, respectively.

El-Wakeil [41] mentioned that in order to produce a botanical insecticide at commercial scale, the source plant biomass has to be available at agricultural scale, with no seasonality if possible, unless the plant is extremely abundant in nature or it is grown already for other purposes. *Moringa oleifera* complies with this sustainability criterion, because Moringa trees are wild trees, but they can also be reproduced by cuttings and seed planting. The trees grow fast; in three months, they can be 3 m tall. When *M. oleifera* is cultivated, the leaves can be harvested at intervals ranging from 35 to 60 days [7].

## 5. Conclusions

The results show *M. oleifera* leaves can be used in mite control as ethanolic extract. The extract caused chronic toxicity on females and impaired hatching of *O. punicae* eggs, leading to lower growth rates. Furthermore, egg-laying and damage were reduced in comparison to the control. Further research is necessary, including the assessment of other *M. oleifera* extracts against *O. punicae* and other species of pest mites, as well as studying the extract’s effect on natural enemies.

## Figures and Tables

**Table 1 insects-12-00476-t001:** Qualitative phytochemical (“+” = present; “−“ = absent) screening of ethanolic extract of *Moringa oleifera* leaves.

Bioactive Compound		Test	Bioactive Compound		Test
Alkaloids	+	Dragendorff’s	Flavonoids	+	Shinoda for flavanone’s
		Sonheschain’s			NaOH at 1% for flavanone’s or Xanthone
Tannins	+	FeCl_3_ for gallic acid	Quinones	+	NH_4_OH for Anthraquinone
		Ferrocyanide for phenols			H_2_SO_4_ for Anthraquinone
Carbohydrates	+	Molisch’s			Bröntraguer’s for benzoquinone
Carotenoids	+	H_2_SO_4_ and FeCl_3_ reagents	Soluble starch	+	KOH and H_2_SO_4_
Sugar reducers	+	Fehling’s	Coumarins	+	Erlich’s
		Benedict’s	Cyanogenic glycosides	+	Grignard’s
Saponins	+	Bouchard for steroidal saponins	Terpenoids	−	Ac_2_O
	−	Foam	Purines	−	HCl
	−	Rosenthaler	Phenols	+	FeCl_3_

**Table 2 insects-12-00476-t002:** Effect of ethanolic extract of *Moringa oleifera* leaves at different concentrations on *Oligonychus punicae* females.

Concentration (%)	Average Mortality (±SE) Percentage *		
	24 **	48 **	72 **
0.1	0.00 ± 0.00 d	3.33 ± 3.33 d	10.37 ± 0.37 d
0.5	3.33 ± 0.33 d	10.00 ± 0.00 c	17.04 ± 2.96 d
1	10.00 ± 0.00 d	16.67 ± 0.33 c	20.37 ± 5.46 cd
5	16.67 ± 3.33 d	23.33 ± 0.33 c	31.11 ± 1.11 c
10	36.67 ± 3.33 c	63.33 ± 0.33 b	72.59 ± 8.12 b
15	53.33 ± 3.33 b	76.67 ± 0.33 ab	89.63 ± 0.37 a
20	70.00 ± 5.77 a	83.33 ± 0.33 a	96.67 ± 3.33 a
LC_50_(CI_95_)	LC_90_(CI_95_)	b ± EE	χ^2^
7.99	15.68	7.50 ± 0.94	63.50 ***
(6.87–8.95)	(13.86–18.63)		

* Mortality corrected using Abbot’s formula (Abbott 1925). ** Means values and ± standard error (SE) is presented. Different letters indicate significant differences. LC: Lethal concentration killing 50(90) % of ABM’s population. CI: Confidence interval at 95%. b: slope± standard error. χ^2^: Chi-square value. *** Level of significance *p* < 0.0001.

**Table 3 insects-12-00476-t003:** Effect of ethanolic extract of *Moringa oleifera* leaves on oviposition and growth rate of *Oligonychus punicae*.

Concentration %(*v*/*v*)	Average Number of Eggs ± SE	OAP (%)	Average Number of Eggs ± SE	OAP (%)	Average Number of Eggs ± SE	OAP (%)	Growth Rate ± SE
	24 h *		48 h		72 h		
Control	61.33 ± 1.533 a		93.33 ± 3.51 a		109.67 ± 1.53 a		0.8267 ± 0.00 a
0.1	58.33 ± 3.06 a	−2.54 ± 2.27	88.00 ± 1.00 b	−2.92 ± 1.31	102.00 ± 2.65 b	−3.63 ± 1.98	0.8000 ± 0.01 b
0.5	47.00 ± 1.00 b	−13.23 ± 0.97	72.67 ± 2.52 c	−12.45 ± 0.37	81.67 ± 1.53 c	−14.64 ± 1.07	0.7300 ± 0.01 c
1	30.00 ± 1.00 c	−34.31 ± 1.14	37.67 ± 1.53 d	−42.46 ± 3.19	46.67 ± 1.53 d	−40.30 ± 1.75	0.5633 ± 0.01 c
5	28.00 ± 1.00 c	−37.30 ± 2.59	31.67 ± 1.53 e	−49.34 ± 0.61	40.33 ± 0.58 e	−46.22 ± 0.39	0.5200 ± 0.00 d
10	4.33 ± 0.58 d	−86.83 ± 1.33	5.33 ± 0.58 f	−89.16 ± 1.45	9.00 ± 1.00 f	−84.83 ± 1.75	0.0500 ± 0.04 d
15	2.33 ± 0.58 d	−92.66 ± 1.89	5.33 ± 0.58 f	−89.16 ± 1.45	9.00 ± 1.00 f	−84.83 ± 1.75	−0.0033 ± 0.03 e
20	1.67 ± 0.58 d	−94.69 ± 1.89	2.67 ± 0.58 f	−94.45 ± 1.19	2.67 ± 0.58 g	−95.26 ± 0.95	−0.4000 ± 0.00 e

* Means values and ± standard error (SE) is presented. Different letters indicate significant differences (*p* < 0.05; ANOVA and Tukey’s HSD test). OAP, percentage of oviposition activity compared to control.

**Table 4 insects-12-00476-t004:** Residual effect of ethanolic extract of *Moringa oleifera* leaves on *Oligonychus punicae* eggs (mean ± SE) (%).

Concentration	Hatched Eggs *	Reduction of Viable Eggs
Control	55.33 ± 1.53 a	
0.1	41.00 ± 2.00 b	−14.90 ± 2.30
0.5	23.33 ± 2.08 c	−42.57 ± 2.93
1	9.67 ± 1.15 d	−70.25 ± 3.57
5	3.67 ± 0.58 e	−87.60 ± 1.64
10	0.00 ± 0.00 f	−100 ± 0.00
15	0.00 ± 0.00 f	−100 ± 0.00
20	0.00 ± 0.00 f	−100 ± 0.00

* Means values and ± standard error (SE) is presented. Different letters indicate significant differences (*p* < 0.05; ANOVA and Tukey’s HSD test).

**Table 5 insects-12-00476-t005:** Effects of ethanolic extract of *Moringa oleifera* leaves on *Oligonychus punicae* feeding rate.

Concentration %(*v*/*v*)	Average Damage (%) ± SE	Food Intake (%) ± SE	Average Damage (%) ± SE	Food Intake (%)	Average Damage (%) ± SE	Food Intake (%) ± SE
	24 h *		48 h		72 h	
Control	12.00 ± 0.58 a		17.00 ± 0.58 a		26.00 ± 0.58 a	
0.1	11.00 ± 0.58 ab	−4.33 ± 0.22	15.33 ± 0.33 a	−5.12 ± 0.95	23.67 ± 0.67 a	−4.71 ± 0.71
0.5	9.33 ± 0.33 b	−12.44 ± 1.27	12.00 ± 0.58 b	−17.24 ± 3.98	20.00 ± 0.58 b	−13.05 ± 2.18
1	5.67 ± 0.33 c	−35.81 ± 3.44	9.33 ± 0.33 c	−29.06 ± 3.08	16.33 ± 0.88 c	−22.89 ± 3.58
5	4.33 ± 0.33 cd	−46.81 ± 4.73	7.00 ± 0.58 cd	−41.67 ± 4.81	13.33 ± 0.67 c	−31.14 ± 2.37
10	2.33 ± 0.33 de	−67.52 ± 3.94	4.67 ± 0.33 de	−57.02 ± 1.59	8.00 ± 0.58 d	−52.97 ± 3.05
15	2.33 ± 0.88 de	−67.94 ± 11.41	3.00 ± 0.58 ef	−70.11 ± 5.29	3.67 ± 0.33 e	−75.25 ± 2.39
20	1.33 ± 0.33 e	−80.16 ± 4.42	1.67 ± 0.67 f	−82.26 ± 6.92	2.33 ± 0.33 e	−83.57 ± 2.15

* Means values and ± standard error (SE) is presented. Different letters indicate significant differences (*p* < 0.05; ANOVA and Tukey’s HSD test).

## Data Availability

The data presented in this study are available on request from the corresponding author.

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
