# Peer review of "Evaluation of Ethanol Extract of Moringa oleifera Lam. as Acaricide against Oligonychus punicae Hirst (Trombidiformes: Tetranychidae)"

_insects, 2021, doi:10.3390/insects12050476_

Round 1
Reviewer 1 Report
Line 2, include authority and family for M. o. Intake should be lower case.
Line 27, delete “the” before humans.
Line 28, delete “work”. Through the manuscript work is not needed following research.
Line 29, add “from” before M.o. Replace “over” with on.
Line 37, delete other.
Line 38, replace minimizing with minimize.
Line 39, delete “work” following research.
Line 40, replace “over” with on.
Line 52, italicize O.
Lines 56 & 57, include accepted common names when available.
Line 87, delete “work” following research.
Line 88, italicize P. d.
Line 93, italicize M. o.
Line 94, replace “at” with in.
Line 96, what is “consistent weight”?
Line 99, why total darkness?
Line 130, was the water tap water, describe.
Line 132, replace “laying” with lying.
Line 134, how did you quantify damage?
Line 216, control values are missing from table 2.
Line 230, add “compared to control”.
Line 251, replace plants with “plant” and add “species”.
Line 252, add reference for previous sentence.
Line 305, begin sentence with “In this research,”
Line 306, italicize M. o.
Line 329, delete “s” from grows.
Check references for italicization of scientific names.
Author Response
Dear Reviewer, the file with the answers to your questions is attached. On the other hand, we accept all your suggestions made to the manuscript.
Please see the attachment

Reviewer 2 Report
Please read the comments and provide explanation to questions directed to the Author/s. Why used only female mites in the experiments? Also in the experiments you used only negative control. Research like this you should have a positive control also that you could compared your results effectively against positive control results.
Simple Summary and abstract is similar.
Table 1 already discussed in the second paragraph of the Introduction section , why Table 1?
Table 2 also discussed in detail in page 5 see under sub heading 3.2. Mortality, oviposition growth rate ..........etc. Also Table 2 caption has no enough information if it is related to mortality, or egg deterrence etc.
I would suggest that statistical analyses should be check by an statistician if properly used and interpreted.

Author Response
Dear Reviewer, the file with the answers to your questions is attached. On the other hand, we accepted all your suggestions made to the manuscript (see the paper).
Regards
Please see the attachment

Round 2
Reviewer 2 Report
The Author/s missed some corrections. I have corrected them using track changes. The manuscript is improved.
